# Reference genes for gene expression analysis in the fungal pathogen *Neonectria ditissima* and their use demonstrating expression up-regulation of candidate virulence genes

Liz M. Florez[1,2], Reiny W. A. Scheper[3], Brent M. Fisher[3], Paul W. Sutherland[4], Matthew D. Templeton[1,2], Joanna K. Bowen[1]*

1 Bioprotection, The New Zealand Institute for Plant & Food Research Limited, Auckland, New Zealand, 2 School of Biological Sciences, University of Auckland, Auckland, New Zealand, 3 Bioprotection, The New Zealand Institute for Plant & Food Research Limited, Havelock North, New Zealand, 4 Food Innovation, The New Zealand Institute for Plant & Food Research Limited, Auckland, New Zealand

* Joanna.bowen@plantandfood.co.nz

## Abstract

European canker, caused by the necrotrophic fungal phytopathogen *Neonectria ditissima*, is one of the most damaging apple diseases worldwide. An understanding of the molecular basis of *N. ditissima* virulence is currently lacking. Identification of genes with an up-regulation of expression during infection, which are therefore probably involved in virulence, is a first step towards this understanding. Reverse transcription quantitative real-time PCR (RT-qPCR) can be used to identify these candidate virulence genes, but relies on the use of reference genes for relative gene expression data normalisation. However, no report that addresses selecting appropriate fungal reference genes for use in the *N. ditissima*-apple pathosystem has been published to date. In this study, eight *N. ditissima* genes were selected as candidate RT-qPCR reference genes for gene expression analysis. A subset of the primers (six) designed to amplify regions from these genes were specific for *N. ditissima*, failing to amplify PCR products with template from other fungal pathogens present in the apple orchard. The efficiency of amplification of these six primer sets was satisfactory, ranging from 81.8 to 107.53%. Analysis of expression stability when a highly pathogenic *N. ditissima* isolate was cultured under 10 regimes, using the statistical algorithms geNorm, NormFinder and BestKeeper, indicated that *actin* and *myo-inositol-1-phosphate synthase* (*mips*), or their combination, could be utilised as the most suitable reference genes for normalisation of *N. ditissima* gene expression. As a test case, these reference genes were used to study expression of three candidate virulence genes during a time course of infection. All three, which shared traits with fungal effector genes, had up-regulated expression *in planta* compared to *in vitro* with expression peaking between five and six weeks post inoculation (wpi). Thus, these three genes may well be involved in *N. ditissima* pathogenicity and are priority candidates for further functional characterization.

**Data Availability Statement:** All relevant data are within the paper and its Supporting Information files.

**Funding:** LMF, RWAS, BMF, PWS, MDT and JKB all received funding from The New Zealand Institute for Plant and Food Research Limited, Strategic Science Investment Fund, Project number: 12070. The funders had no role in study design, data collection and analysis, decision to publish, or preparation of the manuscript.

**Competing interests:** The authors have declared that no competing interests exist.

## Introduction

The filamentous fungus *Neonectria ditissima*, (Tul. & C. Tul.) Samuels & Rossman is the causal agent of European canker (EC) in apple. Although able to infect a wide range of hardwood trees species [1] the disease in apple can be especially destructive with significant economic ramifications in wet, moderate climates mainly due to twig dieback [2]. EC has been recorded in the apple growing regions of North and South America, Europe, Asia and New Zealand resulting in tree loss. In some countries, fruit loss challenges the profitability of production [3–6]. EC early symptoms of infection are reddish-brown lesions around a wound, such as a leaf scar, spur or pruning wound. Over time a canker develops that can ultimately girdle the trunk or branch, causing the death of any distal shoots [7]. This disease occurs predominantly during wet seasons when dispersal of ascospores and conidia, and infection in orchards is facilitated [8]. Ascospores, produced in red perithecia, can be observed within a year after initial canker formation. These two-celled spores can be expelled from the perithecium and wind-dispersed, or exuded as a white-cream sticky mass and splash dispersed, during high humidity periods [9]. In the early stages of canker, conidia are released from white-cream sporodochia and splash dispersed between adjacent trees. Thus, compared to ascospores, conidia can only be locally spread. However, regardless of spore type, spore dispersal, germination and infection are highly facilitated by rainfall.

Control measures for EC focus on inoculum removal through pruning and fungicide application, which work as a temporary protection barrier against fungal ingress. Even when using the most stringent fungicide programmes combined with pruning, the incidence of canker still increases but at a rate that is slower than if no control measures were adopted [4]. A promising direction to realise sustainable control comes from identification of *Malus* x *domestica* germplasm that varies in susceptibility to *N. ditissima* [10,11]. Genetic mapping has identified an *N. ditissima* resistance locus, *Rnd1*, from the cultivar (cv) 'Robusta 5' [12]. SNP markers have been developed for this locus, hinting at the development of an EC-resistant apple cultivar using marker-assisted selection (MAS, [12]).

Although deployment of resistance in germplasm can be effective for disease control, resistance can often be overcome by the pathogen. An understanding of underlying molecular mechanisms and evolutionary forces at play are needed to develop a long-term management strategy to effectively control the disease. Knowledge of the molecular basis of the interaction between *N. ditissima* and apple is very limited. No specific molecular resistance mechanisms have yet been reported and very little is known about the interaction that precedes either symptom expression or successful host defence. Indeed, there is a dearth of knowledge regarding *N. ditissima* virulence mechanisms. Analysing the gene expression of candidate virulence genes in *N. ditissima* during plant infection is the first step towards filling this knowledge gap, with an up-regulation of gene expression during growth *in planta* compared to that *in vitro* suggesting an involvement in virulence.

Although the price of RNAseq experiments is continuously falling, providing a means of accurately analysing gene expression, reverse transcription quantitative real-time PCR (RT-qPCR) remains a valid methodology, especially where detailed time courses of infection are investigated including multiple time points, the scale of which could still render analysis by RNAseq too costly. Indeed, significant progress has been made in understanding plant-fungal interactions since the development of RT-qPCR, due to its sensitivity and ease of use, although it does require rigorous standardisation to accurately interpret the data and generate reliable results [13]. Quantification errors in RT-qPCR data can occur due to variations in RNA concentration, RNA quality, efficiency of cDNA synthesis and PCR amplification. Moreover, in order to allow comparison of expression levels in a disease time course, reference genes are

required to account for differences in fungal biomass between samples and potential differences in total RNA extraction efficiencies.

Typically 'housekeeping' genes (genes required for basic cellular functions) have been used as reference genes for data normalisation with expression independent of the experimental condition [14,15]. However, such genes are still regulated to some extent, reinforcing the opinion that there is no universal reference gene with expression levels that remain constant across all conditions [16,17]. Since even small variations of an internal control can lead to inaccuracies in expression data, it is critical to validate stable expression of reference genes prior to their use for normalisation in RT-qPCR analysis. When validating reference genes, there is a "circular problem" of evaluating the expression stability of a candidate gene when there is no reliable measure available to normalise the candidate. However, statistical algorithms such as geNorm [18], NormFinder [19], and BestKeeper [20] permit a careful selection of a set of genes that display minimal variation across different biological conditions.

To date, a study dedicated to the selection and validation of suitable reference genes in *N. ditissima* has not been reported. Therefore, the purpose of this study was to identify a robust set of reference genes to be used for gene expression profiling in the fungal pathogen *N. ditissima*. A set of housekeeping genes was selected as potential reference genes to be tested for their stability across different growth conditions. As a test case, the most stable reference genes were then applied to quantify expression of candidate virulence genes in *N. ditissima*, during a time course of infection *in planta*.

## Methods

### Fungal material, growth conditions and sampling

The *N. ditissima* pathogenic isolate 23606 from the International Collection of Microorganisms from Plants (ICMP; Manaaki Whenua—Landcare Research, New Zealand; previously referred to as RS324p), collected from a 12-year-old *M.* x *domestica* cv. 'Golden Delicious' tree, New Plymouth, Taranaki, 2009, was used in this study [21]. ICMP 23606 was cultured on a modified version of Matsushima's medium (MM, Matsushima 1961) as adjusted by Dubin and English [22] under near ultraviolet (NUV) light at 20˚C to encourage conidial production. Conidia for either seeding liquid cultures or for use in pathogenicity assessments were collected by washing 4-week-old MM cultures with sterile water (SW, Milli-Q Integral Water Purification System for Ultrapure water, Merck KGaA, MA, USA), then filtered through glass wool to remove mycelial debris. When required, spore concentrations were adjusted with the aid of a haemocytometer.

ICMP 23606 was sub-cultured under 10 different liquid culture conditions for gene expression stability assessment. For a control of vegetative growth under nutrient-rich conditions, the isolate was cultured in Potato Dextrose Broth (PDB, Difco$^{TM}$, NJ, USA). Starvation conditions were realised by culturing in MM which also encouraged spore production. Additives to impose stress were added to PDB and adjusted to pH 6.5. For osmotic stress: 1 M sorbitol or 3 M sodium chloride (NaCl). For cell wall stress: 1% (w/v) aqueous Congo Red to a final concentration of 100 µg/mL or 0.03% (w/v) Calcofluor White, both filter-sterilised through a 0.22 µM filter (Ahlstrom-Munksjö, Helsinki, Finland) prior to adding to PDB. For oxidative stress: 30 mM hydrogen peroxide ($H_2O_2$). For toxic stress: 0.22 µM filter-sterilised caffeine to a final concentration of 2.5 mM. Cold and heat stress were induced by growing *N. ditissima* in PDB at 4˚C and 37˚C, respectively. Liquid cultures that were not exposed to cold and heat stress were incubated at 20˚C. All liquid cultures were incubated for 24 hours at 90 rpm and then filtered through Miracloth (Merck KGaA, MA, USA), by rinsing with SW, to collect the mycelium, which was snap-frozen in liquid nitrogen prior to RNA extraction.

*Botryosphaeria stevensii* RS3 (from the Plant & Food Research Culture Collection, Auckland, New Zealand), *Cladosporium* sp. ICMP 15697, *Colletotrichum acutatum* ICMP 13946, *Colletotrichum gloeosporioides* ICMP 10112, *Neofabraea alba* CBS 518 (from the Centraalbureau voor Schimmelcultures–Westerdijk Fungal Biodiversity Institute, CBS-KNAW, Netherlands), *Neofabraea malicortis* CBS 102863, *Neofabraea perennans* CBS 102869, *Venturia inaequalis* isolate ICMP 1639 [23], *V. inaequalis* isolate MNH120 (ICMP 13258, [24]) and *V. inaequalis* isolate EU-B04 [23] were cultured on Potato Dextrose Agar (PDA, Difco™, NJ, USA) at 20°C, under NUV light, for 7 days.

## Plant material and inoculation

One-year-old dormant *Malus* x *domestica* cv. 'Royal Gala' trees on rootstock 'M793' were inoculated with *N. ditissima*, isolate ICMP 23606, in September 2017. Forty-eight potted 1-year-old trees were arranged in a glasshouse, in a randomised block design, with four replicates, each comprising 12 trees. Of these 12 trees, six were inoculated (five inoculation sites per tree) and six mock-inoculated (three sites per tree), with sampling at six time points. Bud scars were used as inoculation sites and were made by breaking off the buds. Only buds on the main leader were used [12]. The scars were at least 25 cm apart and were inoculated within 2 hrs of being made with 10 μl of $1\times10^5$ spores/mL conidial suspension or 10 μL of SW. After inoculation, the relative humidity in the glasshouse was increased to 100% for three days, and then kept at 75%. The average temperature in the glasshouse was 19°C, with lows of 8.5°C and a high of 32°C, over the duration of the experiment. During the inoculation period, temperatures ranged from 22.5°C to 23.5°C, and 10.5°C to 26.5°C during the period of 100% humidity. For future standardisation of sampling times, thermal time (cumulative daily mean air temperature above a base temperature, units = °C days; [25]) were calculated for each sampling date, assuming a threshold base temperature of 0°C, since the fungus still grows at 1°C [21], and any temperature above 16°C was capped at 16°C, as this threshold was identified as the temperature above which the disease does not increase faster [2,26,27].

Branch samples were taken at three (262°C days), four (359°C days), five (455°C days), six (552°C days), eight (753°C days) and 14 weeks post inoculation (wpi; 1347°C days), with inoculated and control samples taken for each time point. Although four biological replicates (consisting of individual trees) were inoculated with pathogen or water for sampling at each time point, single inoculation sites were randomly selected from only three of the biological replicates for RNA isolation. Tissue samples, approximately 1 cm in length including the bud scars, were halved longitudinally with one half being snap-frozen in liquid nitrogen for RNA extraction and the other fixed for light microscopy. For light microscopy, *in planta* samples from six and 14 wpi were sectioned in 1 μm-thick sections of resin-embedded material and stained in a 0.05% solution of toluidine blue in benzoate buffer (pH 4.4), washed in distilled water, dried, mounted in Shurmount (Triangle Biomedical Sciences, St Louis, MO), and observed using an Olympus Vanox AHBT3 microscope (Olympus Optical, Tokyo).

## RNA extraction

Snap-frozen *in vitro* and *in planta* samples were ground with a pestle and mortar, under liquid nitrogen, to a fine powder and stored at -80°C prior to RNA extraction. RNA was extracted from the *in vitro* samples using the Spectrum™ Plant Total RNA kit (Sigma-Aldrich, MO, USA) according to the manufacturers' instructions. RNA from *in planta* samples was extracted following a modified version of the rapid CTAB extraction procedure proposed by Gambino et al. [28]. Briefly, ground samples (approximately 100mg) were transferred to 900 μl of Extraction Buffer (2% (w/v) hexadecyl(trimethyl)ammonium bromide (CTAB), 2.5% (w/v)

polyvinylpyrrolidne (PVP)-40, 2 M sodium chloride, 100 mM TRIS hydrochloride (Tris-HCl) pH 8.0 and 25 mM ethylenediaminetetraacetic acid (EDTA) pH 8.0) with 18 μL of β-mercaptoethanol, pre-warmed to 65˚C, then briefly vortexed. Samples were incubated at 65˚C for 10 min, then extracted with an equal volume of chloroform:isoamyl alcohol (IAA; 24:1 v/v). Samples were vortexed briefly prior to centrifugation at 11,000 g for 10 min at 4˚C. This chloroform:IAA extraction was repeated prior to precipitating the RNA by the addition of lithium chloride (12 M) to a final concentration of 3M. Samples were incubated on ice for at least 30 min. The RNA was then collected by centrifugation at 21,000 g for 20 min at 4˚C, then re-suspended in 500 μL SSTE buffer (1 M sodium chloride, 0.5% sodium dodecyl sulphate, 10 mM Tris-HCl pH 8.0, 1 mM EDTA pH 8.0), which was pre warmed to 65˚C. The RNA was then extracted using an equal volume of chloroform:IAA as described above. RNA was precipitated from the retrieved aqueous phase by the addition of 0.7 volumes of cold (4˚C) isopropanol. Centrifugation was carried out at 21,000 g for 15 min at 4˚C. The remaining pellet was washed in 500 μL of 70% (v/v) ethanol followed by centrifugation at 21,000 g for 10 min at 4˚C.

The RNA was dried for 10 min in a laminar flow hood, and then re-suspended in 30 μL of UltraPure™ DNase/RNase-Free Distilled Water (Invitrogen™, Thermo Fisher Scientific, MA, USA). RNA was treated with DNase I (InvitrogenTM, Thermo Fisher Scientific, MA, USA) according to the manufacturer's instructions to remove any contaminating gDNA. Absence of gDNA in the samples was confirmed by end-point PCR (see below) using primers specific for *N. ditissima myo-inositol-1-phosphate synthase* (*mips*, **Table 1***)* and *Malus* x *domestica glyceraldehyde-3-phosphate dehydrogenase* (*gapdh*, [29]).

RNA sample concentration and purity was assessed using the DeNovix DS-11 spectrophotometer (DeNovix Inc., Wilmington, DE, USA) and concentration and integrity were assessed using the Agilent RNA 6000 Nano Kit (Agilent Technologies, Waldbronn, Germany) in conjunction with the Agilent 2100 Bioanalyzer software according to the manufacturer's instructions. RNA samples of sufficient integrity, with a RIN value of 7 or more, were used for cDNA synthesis.

**Table 1. Candidate reference genes of *Neonectria ditissima*, primers and their efficiencies when used in qRT-PCR.**

| Gene name | Gene ID and accession number | Abbreviation | Forward Primer | Reverse Primer | Size (bp) | Efficiency (E) | R² value* |
|---|---|---|---|---|---|---|---|
| *Actin* | Ndactin MT040710 | *actin* | CTCTGTTCCAGCCCTCAGTC | TCGGACATCGACATCACACT | 92 | 1.97 | 0.9940 |
| *Beta-tubulin* | Ndbtub MT040711 | *Btub* | TGGAAGTCAAGCACGATGAG | ATGTGCCCCACATCTCTTTC | 90 | 2.07 | 0.9959 |
| *Myo-inositol-1-phosphate synthase* | Ndmips MT040712 | *mips* | TGTTCAACATCTGCGAGGAC | GCCTTCCACTGGATACGAGA | 94 | 1.96 | 0.9895 |
| *Elongation factor thermo unstable* | NdEfTu MT040713 | *EfTu* | GATGCCAGTGGATCTTCACC | TGAGGCTTTGTCGAGTGTTG | 82 | 1.84 | 0.9813 |
| *18s ribosomal RNA adenine methylase transferase* | Nd18sAMT MT040714 | *18sAMT* | TCCGCAAGAACAAGACACTG | ACCATCCTCGATGTCCATGT | 167 | 2.01 | 0.9883 |
| *40S ribosomal protein subunit S8* | NdS8 MT040715 | *S8* | CTCTTACCACCCCTCGAACA | TTCTTCACGTCCTCCTCGAC | 183 | 1.91 | 0.9879 |
| *40S ribosomal protein subunit S27a* | NdS27a MT040716 | *S27a* | TCGACAACGTCAAGTCCAAG | CTTCTTGGGGGTGGTGTAGA | 203 | Not tested | Not tested |
| *Ubiquitin-conjugating enzyme* | NdE2 MT040717 | *E2* | CTCCGACATGGAGAGGAGAG | GAGAGGCCCAGATACCCTTC | 235 | Not tested | Not tested |

* Coefficient of correlation.

## cDNA synthesis

cDNA was synthesised from RNA using the High Capacity cDNA Reverse Transcription kit (InvitrogenTM, Thermo Fisher Scientific, MA, USA) following the manufacturer's instructions for 'cDNA synthesis without RNase inhibitor'. 10 μL of reaction mix was combined with 10 μL (67 ng/ μL) of DNase-treated RNA and cDNA synthesised using the following cycling conditions; annealing at 25˚C for 10 min; extension at 37˚C for 120 min and denaturation at 85˚C for 5 min. Two reactions were carried out; one with the reverse transcriptase enzyme and one without it. Subsequently, an end-point PCR (see below) was carried out to confirm absence of gDNA contamination and successful synthesis of cDNA, using primers specific for *N. ditissima mips* (**Table 1**) and *Malus* x *domestica gapdh* [29]; duplicate reverse transcriptase reactions were pooled if judged successful by these criteria.

## Candidate reference and virulence gene selection, and primer design

Candidate reference genes were selected based on their use for normalisation in other fungal pathosystems: *actin*, *β-tubulin* (*Btub*), *mips*, *thermo-unstable elongation factor* (*EfTu*), *18s ribosomal RNA adenine methylase transferase* (*18sAMT*), 40S *ribosomal protein subunit S8* (*S8*), *40S ribosomal protein subunit S27a* (*S27a*), *ubiquitin conjugating enzyme 2* (*E2*) [15]. For these genes, putative *N. ditissima* ICMP 23606 orthologues were selected utilising BLASTn [30] of the *N. ditissima* ICMP 23606 genome [31] against annotated genes from the *N. ditissima* R09/ 05 genome [32]. Putative identification of orthologues from *N. ditissima* ICMP 23606 was confirmed by utilising BLASTn and BLASTp against the databases of the reference RNA sequences (refseq_rna) and reference protein sequences (refseq_protein) respectively, in NCBI. (accessed November, 2019; **S1 Table**).

To maximise the chance of designing specific primer sets for the candidate reference genes, the predicted *N. ditissima* amplification product sequence from every gene was compared against sequences from 10 apple pathogens (that are associated with *Malus* tissue and/or expected to be found in the apple-growing orchard) in the Reference RNA sequences (refseq_rna) database held at the NCBI (accessed November, 2019) using BLASTn 2.9.0 [30] with a statistically significant expect value of *e*-10 (p = 0.005) as a cut-off. The apple pathogens included were: *B. stevensii* RS3, *Cladosporium* sp. ICMP 15697, *C. acutatum* ICMP 13946, *C. gloeosporioides* ICMP 10112, *N. alba* CBS 518, *N. malicortis* CBS 102863, *N. perennans* CBS 102869, *V. inaequalis* isolate ICMP 1639, *V. inaequalis* isolate MNH120 (ICMP 13258) and *V. inaequalis* isolate EU-B04. Primers were designed to bind to sequences that were dissimilar in *N. ditissima* and the 10 apple pathogens. Primers had melting temperatures between 56.4 and 60.3˚C, lengths between 19 to 21 bp, GC content of 50 to 60%, and amplicon sizes from 82 to 235 bp (**Table 1**).

Candidate virulence genes were selected following screening of the predicted proteome of *N. ditissima* ICMP 23606, using SignalP 4.1 [33], to identify putatively secreted proteins and EffectorP 2.0 [34], to identify putative effectors. Predicted protein products were also analysed with InterProScan 5 [35] to identify protein domains. The Pathogen-Host Interaction database (PHI-base [36]) was screened with the candidate genes to identify any similar genes/protein products involved in virulence in other pathogens. Primers were designed using Primer3 software [37] to be specific to either gDNA or cDNA.

## End-point PCR

End-point PCR was carried out in a 20 μL reaction volume using a final concentration of 0.2 μM primers (forward and reverse), 0.2 mM dNTPs, 10X PCR buffer (200mM Tris-HCl), 2 mM magnesium chloride ($MgCl_2$), one unit of Platinum$^{TM}$ *Taq* DNA polymerase enzyme

(Invitrogen^TM, Thermo Fisher Scientific, MA, USA) and 20 ng of DNA template. The PCR cycling programme consisted of an initial denaturation at 95˚C for 2 min, followed by 40 cycles of 95˚C for 10 sec, 55˚C to 60˚C, depending on primer set, for 30 sec and 72˚C for one min per kb; and a final extension period at 72˚C for 10 min. PCR amplification products were visualised following gel electrophoresis in a 2% (w/v) agarose gel in 1x Tris Acetate-EDTA (TAE) buffer with RedSafe^TM (Intron Biotechnology, SEL, Korea), at the manufacturer's recommended concentration, and the ChemiDoc^TM XRS+ system (Bio-Rad, CA, USA). Estimation of amplification product sizes was made by comparison with the 1kb Plus DNA Ladder (Invitrogen^TM, Thermo Fisher Scientific, MA, USA).

## RT-qPCR

RT-qPCR analysis was carried out using *in vitro* samples to test gene primer efficiency and gene expression stability of candidate reference genes. RT-qPCR master mix was based on a 10μL reaction volume using a final primer concentration of 0.25 μM (forward and reverse), 5 μL of LightCycler® 480 SYBR Green I Master Mix (Hoffmann-La Roche, BSL, Switzerland), 1.5 μL of PCR-grade water and 2.5 μL of cDNA (concentration ranged from 50 pg/μL to 500 ng/μL). RT-qPCR master mix was aliquoted into a 384-well plate manually, with three technical replicates per biological replicate. RT-qPCR analysis was carried out on a LightCycler® 480 instrument (Hoffmann-La Roche, BSL, Switzerland) using the SYBR Green detection system. The RT-qPCR cycling conditions for all analysis consisted of a pre-incubation period of 95˚C for 5 min followed by 40 amplification cycles of 95˚C for 10 sec, 60˚C for 10 sec and 72˚C for 15 sec. This was followed by a melting curve analysis cycle of 95˚C for five sec and 65˚C for 1 min, with cooling at 40˚C for 10 sec. The quantification cycle (Cq) values and associated melting curves from all analyses were recorded using the LightCycler® 480 software 1.5.0.39 (Hoffmann-La Roche, BSL, Switzerland).

RT-qPCR analysis was carried out using *in planta* samples for gene expression analysis during an infection time course. RT-qPCR master mix was based on a 28 μL reaction volume using a final forward and reverse primer concentration of 0.5 μM, 14 μL of LightCycler® 480 SYBR Green I Master Mix (Hoffmannn-La Roche, BSL, Switzerland) and 7 μL of cDNA. A Biomek 2000 workstation (Beckman Coulter, CA, USA) was used to aliquot the RT-qPCR master mix into a 384-well plate, with four technical replicates per biological replicate. The RT-qPCR programme was the same used for the *in vitro* samples. The Cq values and associated melting curves from all analyses were recorded using the LightCycler® 480 software 1.5.0.39 (Hoffmannn-La Roche, BSL, Switzerland).

## Primer specificity test

To confirm specificity, end-point PCR was conducted with the primer sets for each gene using genomic DNA (gDNA) from the 10 apple pathogens (previously mentioned) used in the design of the primers as template with gDNA of *N. ditissima* as a positive control. gDNA was extracted from PDA cultures using the DNeasy Plant Mini Kit (QIAGEN, Hilden, Germany) following the manufacturers protocol. End-point PCR was carried out as described previously.

## Primer efficiency test

For the primer efficiency test, *N. ditissima* cDNA, from *in vitro* samples grown in PDB liquid culture, was used as template. Five samples from a 10-fold dilution series of the cDNA (from 500 ng to 50 pg) were used as separate templates. Three technical replicates were used for each dilution including three non-template control replicates. RT-qPCR analysis was carried out as described previously. Primer efficiency was calculated from the Cq values obtained from the

average of the technical replicates. A slope was generated of the regression between the average Cq values and the log values of each sample dilution. Primer efficiency was calculated for the *N. ditissima* candidate reference and virulence genes.

## Expression stability of candidate reference genes

cDNA samples from *N. ditissima* isolate ICMP 23606 cultured under the 10 different growth treatments were used as template. The RT-qPCR analysis for each candidate reference gene consisted of three biological replicates per growth treatment with three non-template control replicates as a negative control. Three technical replicates were carried out for each sample. RT-qPCR analysis was carried out as described previously. The raw Cq data were used as input in three gene expression stability analysis software programs; geNorm [18], NormFinder [19] and BestKeeper [20]. Moreover, geNorm, NormFinder and BestKeeper algorithms were used to generate relative stability values (RSVs) and comprehensive stability values (CSVs). The RSVs were calculated based on the stability values (SV) of each algorithm [38]. The CSVs, allowing the ranking of the candidate reference genes combining the outputs from the three algorithms, were computed from the geometrical mean (GM) of the RSVs of each candidate reference gene.

## Gene expression analysis

For gene expression analysis, cDNA samples were used to assess the expression of three genes of interest from *N. ditissima* using two stably-expressed reference genes identified in this study. The cDNA samples included those from six time points (week 3, 4, 5, 6, 8, and 14) post inoculation, with water ('mock-inoculated' samples) and *N. ditissima* ('inoculated' samples) having three biological replicates per treatment for each time point. Similarly, cDNA of three *in vitro* samples was also included for expression comparisons of *N. ditissima* candidate virulence genes. The cDNA samples from 6 and 8 weeks post-inoculation were also analysed using the two least stably-expressed reference genes identified in this study. RT-qPCR analysis was carried out as described previously. The raw Cq data were normalised using the delta delta Cq method [39].

## Statistical analysis

One-way analysis of variance (ANOVA) was used to determine overall statistically significant differences in gene expression among all the groups (wpi) with a significance level of $p < 0.05$. For determination of differences in gene expression occurring between specific sets of groups (wpi), a post-hoc Tukey-Honest Significant Difference (HSD) test was used with a significance level of $p < 0.01$. A post-hoc Tukey-HSD test was only run if ANOVA analysis gave a significant p value. Analyses were carried out using the Genstat software 18th Edition (VSN International, 2017).

## Results

### Candidate reference genes

In this study eight *N. ditissima* genes were selected and assessed for suitability as RT-qPCR reference genes (**Table 1** and **S1–S3 Tables**) by screening them across a set of 10 different growth conditions, 9 of which exerted abiotic stress.

**The majority of primers designed for RT-qPCR are specific for *N. ditissima*.** None of the predicted amplification products from *N. ditissima* were identical to the sequences of the eight candidate reference genes from the other apple pathogens present in the NCBI Reference

RNA sequences (refseq_rna) database (**S2 Table**). Conventional end-point PCR demonstrated that the primer sets for amplification of the *actin*, *mips*, *S8* and *18sAMT* sequences were specific and generated amplification products of the expected size only when template from *N. ditissima* was used (**Fig 1** and **S1 Fig**). The use of the *Btub* and *EfTu* primer sets resulted in amplification products of the expected size when template from *N. ditissima* was used (**Fig 1**), but also resulted in amplified products of different sizes from 200 bp to 3 kb. The *E2* and *S27a* primer sets showed evidence of cross-reactivity with the template from at least one of the other apple pathogens giving an amplification product of a similar size to that obtained when template was derived from *N. ditissima* (**Fig 1**) and, thus, were excluded from further evaluation. Additionally, a single peak was observed in melt curve analysis of all primer pairs indicating a single PCR product when used in RT-qPCR (**S2 Fig**).

**The efficiency of all primer sets is suitable for RT-qPCR analysis.** When evaluating primer efficiency (E) and the coefficient of correlation ($R^2$) using the standard curve analysis, the average Cq values ranged from 18.64 (for *actin*) to 23.64 (for *mips*, **S3 Table**) when using undiluted cDNA from *N. ditissima* as template. Primer efficiency data are shown in **Table 1**. Standard curve details for each primer set are detailed in **S5 Table**. All had a satisfactory linear relationship ($R^2 > 0.9813$). The *actin*, *mips* and *18sAMT* primer sets had efficiencies (E) closer to 2, with values of 1.96, 1.97 and 2.01, respectively. The efficiency for the *Btub* primer set displayed a higher value (2.07). The primer sets for *S8* and *EfTu* had lower efficiency values but above 1.8, thus, six candidate reference genes were carried into the gene stability analysis.

**Actin and mips are the most stably expressed genes.** The expression of six *N. ditissima* candidate reference genes was analysed by RT-qPCR following growth under 10 different regimes, including 9 stress conditions. The results showed that the Cq values among all genes ranged from 15.1 to 21.9 (**S3 Fig**). *EfTu* showed the highest expression level, whereas *S8* exhibited the lowest (**S3 Fig**). Diverse results were obtained following analysis of gene expression stability using geNorm, NormFinder and BestKeeper algorithms (**Fig 2**).

Overall, all candidate reference target genes displayed high stability according to geNorm parameters (M ≤ 0.5). geNorm analysis ranked *actin* and *mips* as genes with the highest

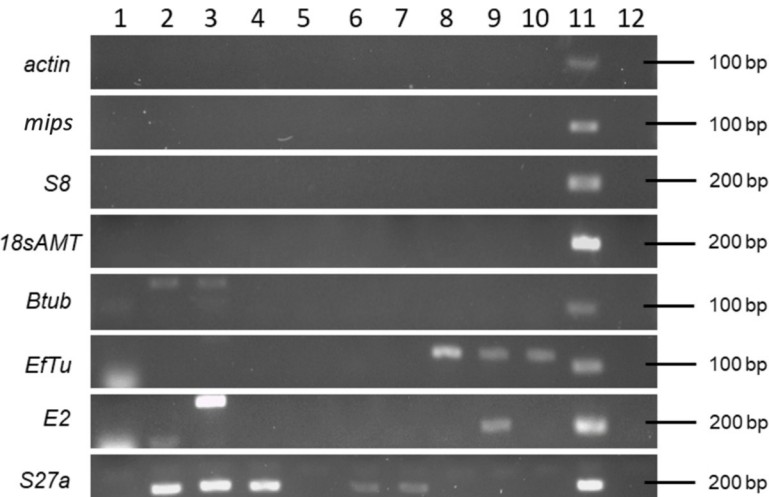

**Fig 1. End-point PCR to test specificity of eight *Neonectria ditissima* candidate reference gene primer sets.**
Amplification using gDNA template from (1) *Botryosphaeria stevensii* RS3, (2) *Cladosporium* sp. ICMP 15697, (3) *Colletotrichum acutatum* ICMP 13946, (4) *Colletotrichum gloeosporioides* ICMP 10112, (5) *Neofabraea alba* CBS 518, (6) *Neofabraea malicortis* CBS 102863, (7) *Neofabraea perennans* CBS 102869, (8) *Venturia inaequalis* 1639, (9) *Venturia inaequalis* MNH120 ICMP 13258, (10) *Venturia inaequalis* EU-B04, (11) *N. ditissima* ICMP 23606 as a positive control and (12) non-template negative control.

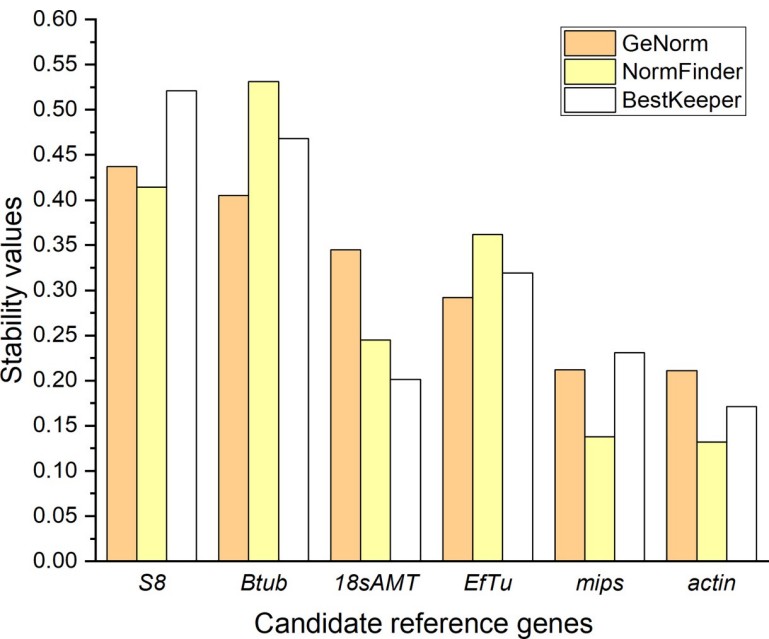

**Fig 2. Stability values of candidate reference genes from *Neonectria ditissima*.** Values based on the three algorithms geNorm (orange), NormFinder (yellow), and BestKeeper (white): *Ribosomal protein S8* (*S8*), *β-tubulin* (*Btub*), *18s ribosomal RNA adenine methylase transferase* (*18sAMT*), *Elongation factor thermos-unstable* (*EfTu*), *myo-inositol-1-phosphate synthase* (*mips*), *actin*.

expression stability based on their small stability values (M = 0.211, 0.212, respectively; **Fig 2** and **Table 2**). geNorm also indicated a low pairwise variation when combining these two genes (V2/3 = 0.088), which is lower than the suggested cut-off threshold of 0.15 that indicates that the geometric mean of these two genes can be used as the optimal normalisation factor in a RT-qPCR analysis (**Fig 3**).

Based on the NormFinder stability value (S), *actin* and *mips* were ranked as the most stably expressed genes with S values below 0.2 (**Fig 2**). *18sAMT*, *EfTu*, *S8* and *Btub* showed less stable expression with S values larger than 0.2 but below 0.6 (**Fig 2**). NormFinder selected *mips* and *EfTu* as the best combination of two reference genes with the lowest S value (0.060, **Table 2**). BestKeeper designated all genes as having satisfactory stability, with no standard deviation

**Table 2. Ranking summary of six *Neonectria ditissima* candidate reference genes using geNorm, NormFinder and BestKeeper.**

| | geNorm | | NormFinder | | BestKeeper | |
|---|---|---|---|---|---|---|
| | Gene | M value | Gene | S value | Gene | SD |
| 1 | *actin* | 0.211 | *actin* | 0.132 | *actin* | 0.171 |
| 2 | *mips* | 0.212 | *mips* | 0.138 | *18sAMT* | 0.201 |
| 3 | *EfTu* | 0.292 | *18sAMT* | 0.245 | *mips* | 0.231 |
| 4 | *18sAMT* | 0.345 | *EfTu* | 0.362 | *EfTu* | 0.319 |
| 5 | *Btub* | 0.405 | *S8* | 0.414 | *Btub* | 0.468 |
| 6 | *S8* | 0.437 | *Btub* | 0.531 | *S8* | 0.521 |
| | *actin* & *mips* [a] | | *mips* & *EfTu* [b] | | *actin* & *18sAMT* [c] | |

[a] Optimum pair of reference genes based on the average pairwise variation V (V2/3 = 0.088).

[b] Optimum pair of reference genes based on stability value (S = 0.06).

[c] Optimum pair of reference genes based on highest correlation (r = 0.991, p < 0.001).

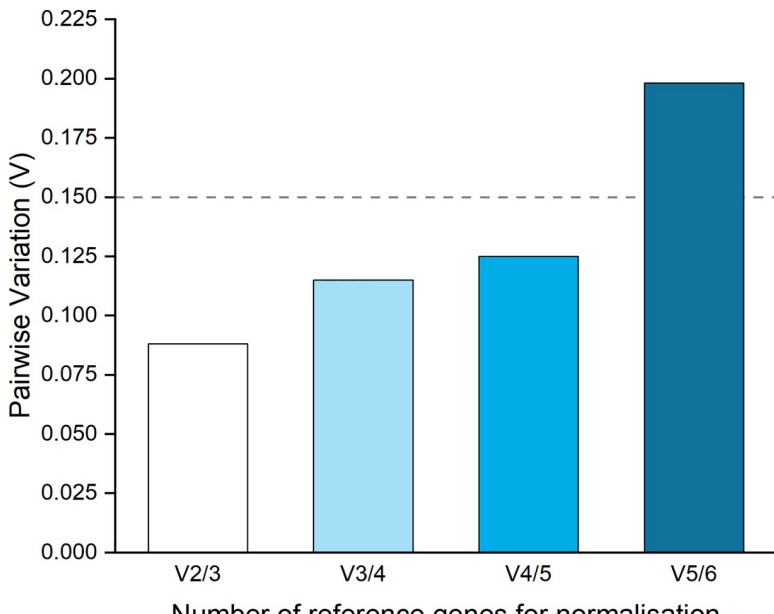

**Fig 3. Determination of the optimal number of reference genes by geNorm software.** V2/3 represents the pairwise variation between the two most stably expressed genes according to geNorm, *actin* and *mips*. V3/4 represents the variation adding third and fourth place, *EfTu* and *18sAMT*. V4/5, includes *18sAMT* and *Btub*. V5/6, includes *Btub* and *S8*.

(SD) values greater than 1.0 (i.e. two-fold change). *actin* and *18sAMT* were ranked according to BestKeeper as the most stably expressed genes with SD values of 0.171 and 0.201, respectively (**Fig 2**), and the best combination of genes to use based on their high correlation (r = 0.991, p < 0.001, **Table 2**). *EfTu* and *mips* were ranked second with SD values less than 0.4 and *Btub* and *S8* had the lowest rank with SD values higher than 0.4 but lower than 0.6 (**Fig 2**).

Due to each software program suggesting a different combination of genes to be used as references in a RT-qPCR analysis (**Table 2**), the output data from the three programs were used to generate comprehensive stability values (CSVs) based on individual stability values (SVs) from each algorithm. The comprehensive rank based on the CSV values indicated that *actin* and *mips* are the most stable genes in *N. ditissima* displaying relatively stable expression patterns under all conditions assessed (**Table 3**).

**Table 3. The relative (RSV) and comprehensive (CSV) stability values of six *Neonectria ditissima* candidate reference genes.**

| | geNorm | | NormFinder | | BestKeeper | | Comp. Rank[2] | |
|---|---|---|---|---|---|---|---|---|
| | Gene | RSV[1] | Gene | RSV | Gene | RSV | Gene | CSV[3] |
| 1 | *actin* | 1.000 | *actin* | 1.000 | *actin* | 1.000 | *actin* | 1.0000 |
| 2 | *mips* | 1.004 | *mips* | 1.045 | *18sAMT* | 1.175 | *mips* | 1.1233 |
| 3 | *EfTu* | 1.039 | *18sAMT* | 1.856 | *mips* | 1.351 | *18sAMT* | 1.4164 |
| 4 | *18sAMT* | 1.302 | *EfTu* | 2.742 | *S8* | 1.865 | *EfTu* | 1.7453 |
| 5 | *Btub* | 1.441 | *S8* | 3.136 | *EfTu* | 2.737 | *S8* | 2.4586 |
| 6 | *S8* | 1.555 | *Btub* | 4.023 | *Btub* | 3.047 | *Btub* | 2.5129 |

[1]RSV: Relative stability value

[2]Comp. Rank: Comprehensive rank

[3]CSV: Comprehensive stability value.

## Candidate virulence genes

The optimised reference genes were used in a test case of *in planta* gene expression of three candidate virulence genes (**Table 4** and **S4 Table**).

The candidate virulence genes (*g4542*, *g5809* and *g7123*), selected are single copy genes, predicted to encode effectors, in that the protein products are small, predicted to be canonically secreted (with the presence of a signal peptide as identified by SignalP 4.1), and are predicted by the algorithm EffectorP [34] to be an effector with probability values of 0.873, 0.827, 0.856, respectively (**S3** and **S4 Tables**). No domains, in addition to the signal peptide, were identified when the predicted protein products were analysed by Interproscan (**S3 Table**). No genes or proteins similar to *g4542* and its predicted encoded product were identified in the public domain. However, a gene encoding a hypothetical protein most similar to *g5809* was found in *Phialemoniopsis curvata* and a protein most similar to the putative product of *g5809* was found in *Scedosporium apiospermum* (both belonging to the Class Sordariomycetes [40,41]). Moreover, a hypothetical gene and protein most similar to *g7123* were identified in *Fusarium oxysporum*. None of the genes were similar to characterised virulence determinants in PHI-base, thus they could be considered *N. ditissima*-specific gene sequences with a potential virulence function, sharing characteristics with many fungal effectors.

## Infection progress across the time course

When assessing *in planta* phenotype after inoculation, the first external canker symptoms were observed at inoculation sites six wpi. Obvious canker symptoms were visible seven wpi. Internal browning of tissue was apparent at six wpi. Following sectioning, at six wpi, fungal hyphae were visible externally at the inoculation point and extended approximately 50 μm in to the tissue, with some initial disruption of the plant cell wall. Later in the infection time course (at 14 wpi) fungal hyphae were clearly visible extending from the wound, accompanied by extensive plant tissue degradation (**Fig 4**).

**Candidate effector genes expression is upregulated *in planta* versus *in vitro*.** Expression of the three N. *ditissima* candidate virulence genes (*g4542*, *g5809*, *g7123*) *in planta* at all the time points post-inoculation was significantly different to their expression *in vitro* (p < 0.001, **Fig 5**).

Expression of *g4542* was up-regulated at three, four, five and six wpi (57-, 63-, 68- and 66-fold, respectively) and was significantly different from its *in vitro* expression (3-fold, p < 0.001, **Fig 5**). A significant decrease at eight and 14 wpi (22- and 14-fold, respectively, p = 0.0091) was observed, however this low expression during the late stages of infection was still significantly greater than *in vitro* expression (p = 0.0058). Expression of *g5809* was up-regulated at three wpi (42-fold), with increases in the following weeks (four wpi, 86-fold and five

**Table 4. Primer sequences of *Neonectria ditissima* candidate virulence genes for qRT-PCR analysis.**

| Gene name | Gene ID and accession number | Abbreviation | Forward Primer | Reverse Primer | Size (bp) | Efficiency (E) | R² value* |
|---|---|---|---|---|---|---|---|
| Candidate effector gene *g4542* | Nd_g4542 MT040718 | *g4542* | GCGGCTTTGTGTGACTATGG | AGATATTGCCTCCCCAAGCT | 148 | 2.02 | 0.9954 |
| Candidate effector gene *g5809* | Nd_g5809 MT040719 | *g5809* | CTCGGTATTGGCCAGACTCA | AGCCAGACCATCTCCCAAC | 127 | 1.92 | 0.9948 |
| Candidate effector gene *g7123* | Nd_g7123 MT040720 | *g7123* | GAATGGTGAGGGTTGGGAGT | AGTTGATAGACCCGGTGCAA | 143 | 1.94 | 0.9951 |

* Coefficient of correlation.

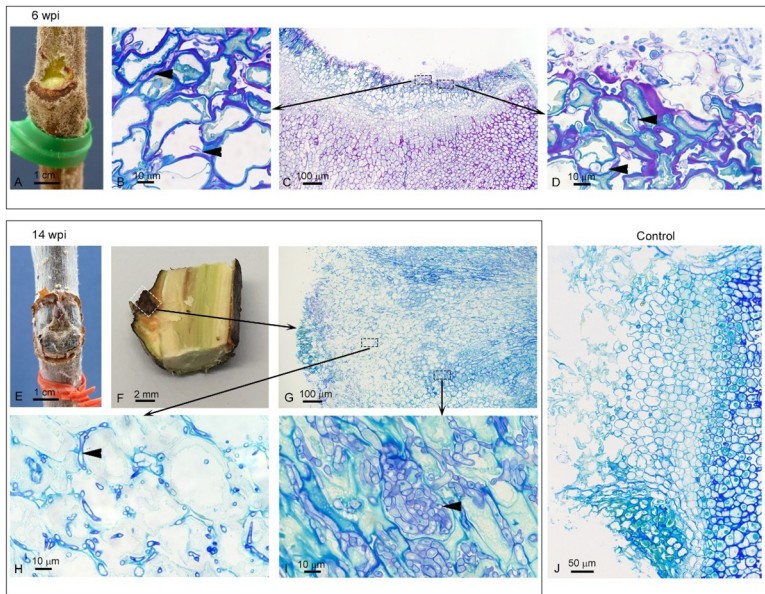

**Fig 4. Symptoms caused by *Neonectria ditissima* on apple (cv. 'Royal Gala') twigs.** Observation of one representative biological replicate from six (A-D) and 14 (E-I) weeks post inoculation (wpi). Six wpi—A: External canker symptoms. C: 1 µm-thick section of resin-embedded material at the entrance point of infection. Black-dashed rectangles indicate close-ups in B and D. B and D: Fungal hyphae (arrow heads: purple staining) penetrating through the tissue. 14 wpi—E: External canker symptoms. F: Longitudinal section through apple twig at inoculation point (white dashed rectangle). G: 1 µm-thick section of resin-embedded material at the entrance point of infection (close-up of area enclosed by white dashed rectangle in F). Black-dashed rectangles indicate close-ups in H and I. H: Plant cell wall break down (pale green staining) as fungal hyphae (arrow heads: purple staining) ramify through the tissue, associated with the apoplast. I: Significant fungal growth (arrow heads: purple staining). J: Control (14 wpi) with no evidence of fungal infection.

wpi, 83-fold), reaching a significant peak of expression by six wpi (271-fold, p < 0.001). By eight wpi (75-fold), and 14 wpi (39-fold), *g5809* showed a decrease in expression significantly different from six wpi (p < 0.001), but similar to that recorded during the initial stages of infection (p = 0.88). At all time points, *g5809* expression *in planta* was significantly greater that its expression *in vitro* (5-fold, p = 0.0076, **Fig 5**). A similar pattern was observed for *g7123*, where gene expression was upregulated at all time points of infection and this was significantly different from its *in vitro* expression (p = 0.0083, **Fig 5**). Up-regulation was noticed at three wpi (38-fold), four wpi (72-fold) with a significant increase by five wpi (111-fold, p = 0.0068). *g7123* expression started decreasing by six wpi (77-fold), with a similar level to that recorded at four wpi (p = 0.65). Significant *g7123* expression reduction was observed late in the infection time course at eight wpi (19-fold, p = 0.0084) and 14 wpi (14-fold, p = 0.0097), however it remained significantly different from expression *in vitro* (3-fold, p = 0.0082). There was no amplification of fungal sequences during RT-qPCR analysis when mock-inoculated cDNA was used as template throughout the time course.

Overall, the patterns of expression of the three candidate virulence genes were very similar when either the combination of the most stable reference genes *actin* and *mips*, or the two least stably expressed genes, *S8* and *Btub*, were used for normalisation. However, the relative expression values were greater when *actin* and *mips* were used for normalisation. Thus, when *S8* and Btub were used, differences in expression of the candidate virulence genes at different time points were deemed to be insignificant (p = 0.1231 to 0.6586, S4 Fig), where with *actin* and *mips* differences were significant (p < 0.001, S4 Fig). Moreover, expression *in planta* versus *in*

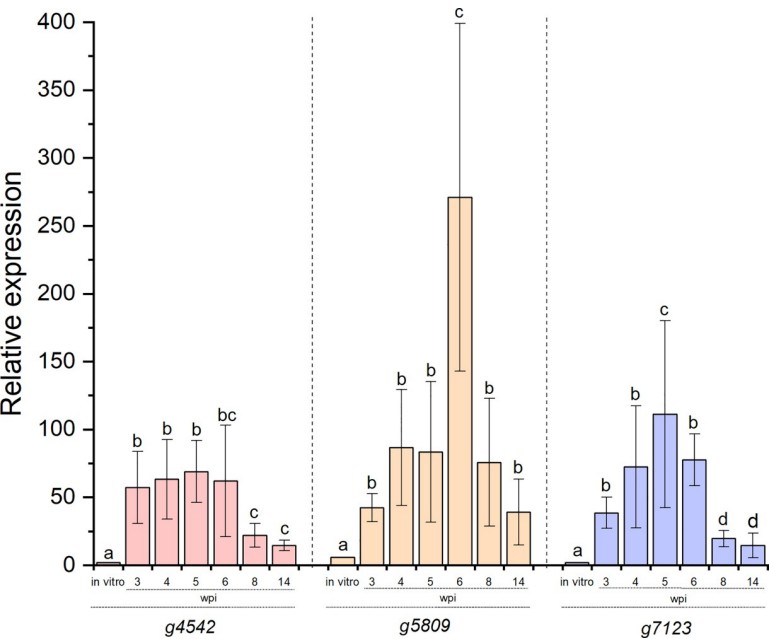

**Fig 5. The relative expression of three *Neonectria ditissima* candidate virulence genes during *in planta* infection.**
*g4542*, *g5809* and *g7123* relative expression *in vitro* versus *in planta* during a time course of infection. wpi: weeks post-inoculation. Error bars represent standard error of the mean (SEM) among biological replicates. For each gene, letters indicate significant differences between time points at p < 0.01. Relative expression was measured through data normalisation using *actin* and *mips* as the reference gene combination.

*vitro* was not significantly different in *g5809* (p = 0.5477) and *g7123* (p = 0.5980) when using *S8* and *Btub*, whereas a significant difference can be observed when using *actin* and *mips* (p < 0.001, S4 Fig).

## Discussion

Suitable reference genes for gene expression analysis in *N. ditissima* have been identified. The first step in the validation of reference genes for *N. ditissima* expression profiling by RT-qPCR required analysis of specificity. A specificity test may not be required when analysing, for example, tissue-cultured plants inoculated with a specific fungus, which have no contaminants. However, replication of natural canker infections, for analysis of pathogenicity, requires the use of mature host material, typically derived from an orchard or glasshouse, hence the requirement for highly specific primer sets. Indeed, in the test case of *in planta* gene expression, samples for the infection time course came from a glasshouse experiment, with the apple material originating from an open hardstand which had the potential of carrying contaminating organisms that are commonly found in orchards. Two of the eight primer sets for the candidate genes (*S27a* and *E2*) lacked specificity, even after annealing temperature adjustments in end-point PCR, showing cross-reactivity with the DNA of other apple pathogens. These primer sets were excluded from further analysis. Moreover, the use of *Btub* and *EfTu* primer sets resulted in non-specific amplification in other organisms, however these primer sets were not initially discarded from further analysis since the non-specific amplified products were not the same size as the amplicon for *N. ditissima*, and thus, during RT-qPCR analysis could be distinguished if amplified. Indeed, during the RT-qPCR assays only single amplification products were observed during the melt-curve analysis (S1 Fig).

When defining the optimal reference genes for a RT-qPCR assay, gene expression stability is a decisive factor. Therefore, the software packages geNorm, NormFinder and BestKeeper were used to provide a reliable measure of gene expression stability. These algorithms have been broadly utilised for validation of reference genes in plant-fungal interactions; such as in cereal-*Fusarium graminearum* [42], rice-*Magnaporthe oryzae* [43], oil palm-*Ganoderma* [44], wheat-rust [16] and sugarcane-*Sporisorium scitamineum* [38] interactions. geNorm software ranked *actin* and *mips* as the most stable genes with the smallest M values, but even genes such as *S8* and *Btub* that had the lowest ranking, had M values below the 0.5 cut-off suggested by the software (**Fig 2**). This threshold is somewhat arbitrary. Indeed, putative reference genes of cucumber with M values less than 1.5 were considered stably expressed [45]. Similarly, a study carried out in *M. oryzae* defined $M \geq 1$ as the cut-off for a gene to be defined as unstable [43]. Therefore, according to geNorm software, all six candidate genes assessed are suitable as reference genes.

The ranking given by NormFinder was very similar to geNorm, with *actin* and *mips* ranked as the most stable genes. The S values obtained in this study (0.132 to 0.531) are similar to the S values obtained when *F. graminearum* reference genes were analysed under stress conditions (0.144 to 0.616, [42], unsurprising given that *Neonectria* and *Fusarium* are closely related. *mips* and *EfTu* were selected as the best combination of two reference genes using NormFinder, excluding the most stable gene *actin*. This result can be explained when observing the Cq data where the values for *mips* and *EfTu* were similar (18 to 20) but different to the *actin* values (16 to 17, **S2 Fig**).

BestKeeper identified *actin* and *18sAMT* as the two most stably expressed genes with the lowest variation (SD values of 0.171 and 0.201, respectively) based on the pair-wise correlation analysis of all pairs of candidate reference genes. Pfaffl et al. [20] considered the expression of any gene with an SD higher than one to be inconsistent which, similar to geNorm, indicated that all six candidate reference genes chosen in this study are stably expressed. However, the outputs from BestKeeper differed to those from the other two software programs. This is a pattern that has been observed in previous studies [15,38,46]. The best combination of two genes to use as references differed among the analyses conducted using the three programs, thus, reinforcing that a comprehensive rank is needed for a clearer consensus. Thus, a comprehensive rank, based on the software output values converted to relative stability values, indicated *actin* and *mips* as the most stably expressed genes in *N. ditissima*.

Previously, actin has been studied extensively for its use as a reference gene for data normalisation in many species [15,42–44] due to its conserved function in cytoskeleton assembly. Even though its expression can vary across different fungal (and plant) species, and experimental conditions, it has proven to be a stable gene to be used for gene expression analysis in *N. ditissima*. Mips is an enzyme known to play a crucial role in cellular structure serving to synthesise a precursor of inositol which is a key component of cellular membranes [47,48]. Its study has been mainly focused on its contribution to resistance towards abiotic and biotic stress and regulation of growth in plants [49–51], oxidative stress in bacteria [52] and regulation of cell growth, structure and intracellular signalling in yeast and fungi [53,54]; however, in this study, *mips* has shown stable expression when *N. ditissima* is grown under stress conditions making it an ideal reference gene for data normalisation. This study has identified, for the first time, a set of reference genes in *N. ditissima*, where *actin*, *mips*, or their combination can be recommended as a reliable tool for normalisation of the expression of genes of interest, facilitating gene expression analysis in the apple–*N. ditissima* interaction.

Three *N. ditissima* genes (*g4542*, *g5809*, *g7123*) were selected in this study for their potential role in *N. ditissima* virulence. *g4542*, *g5809* and *g7123* candidate virulence genes showed evidence of upregulation peaks, compared to expression *in vitro*, at different time points of

infection. *N. ditissima* progresses slowly over weeks rather than hours or days, even when in a conducive environment [55], as used during the time course experiment. *g4542* was upregulated during the early stages (three to six wpi) and *g5809* and *g7123* were mainly upregulated at six wpi and five wpi, respectively. Upregulation of expression of *g4542* and *g7123* compared to *in vitro* ceased at six wpi and *g5809*, at eight wpi. The roles of these predicted virulence genes are currently unknown, although the predicted encoded products of all three share traits with fungal effectors in that they are small, secreted proteins, and have been predicted to be an effector by the algorithm EffectorP. Effector genes often have tailored expression profiles, altering expression depending on their function, the stage of infection and their targets in the plant [56,57]. In the late stages of infection, the expression of the three candidate effector genes was low. However, it would be inaccurate to consider that at 14 wpi, the expression of these genes is down-regulated. In contrast, at 14 wpi, the three candidate effector genes still show relative expression values, compared with expression *in vitro*, from 14-fold to 39-fold induction.

When expression levels of the candidate virulence genes at two time points were analysed using either the two least-stably expressed genes, *S8* and *Btub*, or *actin* and *mips*, the patterns of gene expression were similar, although relative expression levels were lower when *S8* and *Btub* were used. This is unsurprising since although they were the least-stably expressed, *S8* and *Btub* were classified as suitable to be used as reference genes by all the software packages. This phenomenon has also been observed in other studies with a decrease in expression when lower stability genes are used for normalisation [58,59]. However, more dramatic differences in expression levels can also be observed with significant alterations of relativities depending on the reference genes used [60]. Although patterns of expression were retained, statistical analysis of the differences between the relative expression levels was affected, with a decrease in the significance of the differences between expressions at different time points *in planta* compared with expression *in vitro*. This indicates that use of the sub-optimal genes for normalisation may lead to nuanced differences in gene expression being overlooked.

The RT-qPCR data demonstrate that *g4542*, *g5809* and *g7123* expression is likely to be required when *N. ditissima* is infecting apple woody stem tissue, but not during growth *in vitro*, and that they appear to be important during the pre-symptomatic through to the symptomatic phase of the infection. This result suggests that, in common with other studies, the traditional way of looking at necrotrophs as pathogens that do not have an intimate interaction with their host but instead only release toxic molecules and lytic enzymes to decompose plant tissue for nutrition is too simplistic [61], and that *N. ditissima* effectors may well be involved in a nuanced interaction with apple [62]. Indeed, necrotrophic pathogens can rely on effectors that require an interaction with a host susceptibility protein. When this susceptibility protein is the product of a plant resistance (*R*) gene [63], recognition of the effector by the susceptibility protein initiates a cell death response that, rather than restricting the pathogen, as would occur for a biotroph, is of benefit to a necrotroph i.e. an inverse gene-for-gene interaction [64,65]. For example, the wheat *Tsn1* gene encodes a serine/threonine protein kinase-nucleotide binding site-leucine rich repeat domain containing gene, domains characteristic of *R* genes, which recognises the effector ToxA from both *Pyrenophora tritici-repentis* and *Parastagonospora nodorum* to confer susceptibility [66,67]. Alternatively necrotrophic effectors can target a susceptibility protein involved in plant cell metabolism to enhance virulence. For example, the small secreted protein, SsSSVP1, from *Sclerotinia sclerotiorum*, manipulates plant energy metabolism to facilitate infection by targeting a protein component of the plant mitochondrial respiratory chain [68]. Moreover, necrotrophic effectors may suppress initial defence responses; the integrin-like protein SsITL, from *S. sclerotiorum*, suppresses host immunity at the early stage of infection [69] by targeting a chloroplast-localised calcium-sensing receptor to inhibit salicylic acid accumulation [70]. Functional characterisation of the

three candidate effector genes will determine their role in virulence and if they have susceptibility targets or proteins, which act in an inverse gene-for-gene manner that could be removed from the germplasm to enhance resistance to *N. ditissima*.

## Conclusions

Reference genes for RT-qPCR have been identified for the first time in *N. ditissima*. Therefore, RT-qPCR using the validated reference genes will enable a finer dissection of the temporal expression patterns of candidate genes in future studies to enable prioritisation of targets for functional characterisation. Gene expression analysis of three *N. ditissima* candidate virulence genes (*g4542*, *g5809* and *g7123*) provided evidence of significant up-regulation during apple infection, making them good candidates for further functional characterisation to elucidate their role in *N. ditissima* pathogenicity.

## Supporting information

**S1 Fig. Fig 1 raw gel images.**
(PDF)

**S2 Fig. Melt curve analysis for reference genes following RT-qPCR.** Single peaks were observed at the melting temperature (˚C) of the respective amplicons. *actin*—81.6, *Btub*—80.9, *mips*—82.7, *EfTu*—83.8, *S8*–88.0, *18sAMT*—88.5.
(TIF)

**S3 Fig. RT-qPCR Cq values and interquartile ranges for *Neonectria ditissima* candidate reference genes under 10 growth conditions.** Data derived from three technical replicates from three biological replicates.
(TIF)

**S4 Fig.** The relative expression of three Neonectria ditissima candidate virulence genes using (A) the least stable reference genes S8 and Btub versus (B) the most stably expressed reference genes actin and mips. Data derived from three technical replicates from three biological replicates.
(TIF)

**S1 Table. Functional similarities of candidate reference and virulence genes revealed by BLASTn and BLASTp searches against the *Neonectria ditissima* R09/05 genome database in the MycoCosm website.**
(DOCX)

**S2 Table. Functional similarities of candidate reference and virulence genes revealed by BLASTn and BLASTp searches against the databases of Reference RNA sequences (refseq_rna) and reference protein sequences (refseq_protein) respectively, in NCBI.**
(DOCX)

**S3 Table. Potential protein domains found in the candidate reference and virulence genes when searched with InterProScan5.**
(DOCX)

**S4 Table. Details of candidate virulence genes in *Neonectria ditissima*.**
(DOCX)

**S5 Table. Primer efficiency when amplifying candidate reference genes.**
(DOCX)

## Acknowledgments

The authors would like to thank Shamini Pushparajah and Brogan McGreal for providing the gDNA of *N. ditissima* and the remaining apple pathogens used in the gene specificity analysis. In addition they would like to thank Kar Mun Chooi and Mark Andersen for critical reading of the manuscript.

## Author Contributions

**Conceptualization:** Reiny W. A. Scheper, Matthew D. Templeton, Joanna K. Bowen.

**Formal analysis:** Liz M. Florez.

**Investigation:** Liz M. Florez, Reiny W. A. Scheper, Brent M. Fisher, Paul W. Sutherland.

**Methodology:** Reiny W. A. Scheper, Joanna K. Bowen.

**Supervision:** Matthew D. Templeton, Joanna K. Bowen.

**Visualization:** Liz M. Florez, Paul W. Sutherland.

**Writing – original draft:** Liz M. Florez, Joanna K. Bowen.

**Writing – review & editing:** Liz M. Florez, Reiny W. A. Scheper, Matthew D. Templeton, Joanna K. Bowen.

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
