## [Decision Letter · Decision Letter 0]

17 Sep 2020

PONE-D-20-24931

Reference genes for gene expression analysis in the fungal pathogen *Neonectria ditissima* and their use demonstrating expression up-regulation of candidate virulence genes

PLOS ONE

Dear Dr. Bowen,

Thank you for submitting your manuscript to PLOS ONE. After careful consideration, we feel that it has merit but does not fully meet PLOS ONE’s publication criteria as it currently stands. Therefore, we invite you to submit a revised version of the manuscript that addresses the points raised during the review process.

We look forward to receiving your revised manuscript.

Kind regards,

Raffaella Balestrini

Academic Editor

PLOS ONE

Journal Requirements:

Reviewers' comments:

Reviewer's Responses to Questions

**Comments to the Author**

1. Is the manuscript technically sound, and do the data support the conclusions?

Reviewer #1: Yes

Reviewer #2: Yes

2. Has the statistical analysis been performed appropriately and rigorously? 

Reviewer #1: Yes

Reviewer #2: Yes

3. Have the authors made all data underlying the findings in their manuscript fully available?

Reviewer #1: Yes

Reviewer #2: Yes

4. Is the manuscript presented in an intelligible fashion and written in standard English?

Reviewer #1: Yes

Reviewer #2: Yes

5. Review Comments to the Author

Reviewer #1: The authors reported the identification of reference genes for RT-qPCR analysis in the fungal pathogen N. ditissima. The work has been carried out in a correct and “classical” way for this type of manuscript and reports interesting and useful results for researchers dealing with this pathogen.

I suggest some improvements:

In M&M please specify how the biological replicates were obtained.

In the results, I suggest separating the sections relating to the reference genes from the virulence genes. The paragraph starting on line 281 can be confusing for the reader. The authors can begin to describe the reference genes and the data for their selection, and at the end of the results they can introduce the virulence genes and their quantification.

It would be interesting to compare in Fig 5 how the expression levels of the virulence genes change if they are normalized with the worst combination of reference genes. It would be interesting to show that the values are very different.

Throughout the manuscript please change qRT-PCR to RT-qPCR.

Reviewer #2: The paper from Florez and co-authors present the identification of housekeeping genes and the consequent utilisation of them to analyse the expression profile of three candidate effectors.

The paper is well written and well presented, the objective of the work is clearly stated and it is easy to follow the workflow. The data are statistical supported and the conclusions are in line with the result.

It was a pleasure to read such a well-organized paper, with clear a materials and methods section and no speculation the results obtained.

6. PLOS authors have the option to publish the peer review history of their article (what does this mean?). If published, this will include your full peer review and any attached files.

Reviewer #1: No

Reviewer #2: No

---

## [Author Response · Author response to Decision Letter 0]

15 Oct 2020

In response to the Journal requirements we can confirm that:

• our manuscript meets PLOS ONE’s style requirements, including those for file naming.

• we are providing an original uncropped and unadjusted gel result file relating to the composite figure Fig 1 in the results named ‘S1_Fig’ which can be found in Supporting Information. The other supporting files have been renumbered.

We have addressed the comments from the Reviewers:

In M&M please specify how the biological replicates were obtained. 

We have included (lines 128-132) the statement: One-year-old dormant Malus x domestica cv. ‘Royal Gala’ trees on rootstock ‘M793’ were inoculated with N. ditissima, isolate ICMP 23606, in September 2017. Forty-eight potted 1-year-old trees were arranged in a glasshouse, in a randomised block design, with four replicates, each comprising 12 trees. Of these 12 trees, six were inoculated (five inoculation sites per tree) and six mock-inoculated (three sites per tree), with sampling at six time points.

And at lines 145-147: Although four biological replicates (consisting of individual trees) were inoculated with pathogen or water for sampling at each time point, single inoculation sites were randomly selected from only three of the biological replicates for RNA isolation.

In the results, I suggest separating the sections relating to the reference genes from the virulence genes. The paragraph starting on line 281 can be confusing for the reader. The authors can begin to describe the reference genes and the data for their selection, and at the end of the results they can introduce the virulence genes and their quantification.

In the Results section, we have separated the results pertaining to reference gene selection from analysis of candidate virulence gene expression, so it is clearer for the reader. The virulence gene analysis starts once the results from the selection of reference genes is completed. For example the section originally from lines 289-302 has been moved to lines 381-395. The Results section is now split in to two; the first section is on the candidate reference genes with new subheadings within this section, and the second section is on candidate virulence genes, again with new subheadings.

It would be interesting to compare in Fig 5 how the expression levels of the virulence genes change if they are normalized with the worst combination of reference genes. It would be interesting to show that the values are very different.

We proceeded to do the comparison of the gene expression analysis of the virulence genes utilising the worst reference genes (S8 and Btub) at two time points (week 6 and week 8 versus in vitro) for comparison with the data analysed using the two recommended reference genes. Using S8 and Btub for normalisation results in a similar expression pattern of the candidate virulence genes to when actin and mips are used. This result is not unexpected since, for example, the geNorm M-value threshold analysis indicates that S8 and Btub pass the M-value threshold for gene stability and so are still suitable for normalisation. However, the relative expression values are lower when the least stable reference gene combination is used and affects the statistical analysis of the significance of the differences between expression at different time points. This comparison has been added to the Methods, Results and Discussion sections as follows:

Methods line 273-274: The cDNA samples from 6 and 8 weeks post-inoculation were also analysed using the two least stably-expressed reference genes identified in this study.

Results line 445-452: Overall, the patterns of expression of the three candidate virulence genes were very similar when either the combination of the most stable reference genes actin and mips, or the two least stably expressed genes, S8 and Btub, were used for normalisation. However, the relative expression values were greater when actin and mips were used for normalisation. Thus, when S8 and Btub were used, differences in expression of the candidate virulence genes at different time points were deemed to be insignificant (p = 0.1231 to 0.6586, S4 Fig), where with actin and mips differences were significant (p < 0.001, S4 Fig). Moreover, expression in planta versus in vitro was not significantly different in g5809 (p = 0.5477) and g7123 (p = 0.5980) when using S8 and Btub, whereas a significant difference can be observed when using actin and mips (p < 0.001, S4 Fig). 

We have included a further figure in Supporting Information:

S4 Fig. The relative expression of three Neonectria ditissima candidate virulence genes using (A) the least stable reference genes S8 and Btub versus (B) the most stably expressed reference genes actin and mips. Data derived from three technical replicates from three biological replicates.

Discussion line 524-535: When expression levels of the candidate virulence genes at two time points were analysed using either the two least-stably expressed genes, S8 and Btub, or actin and mips, the patterns of gene expression were similar, although relative expression levels were lower when S8 and Btub were used. This is unsurprising since although they were the least-stably expressed, S8 and Btub were classified as suitable to be used as reference genes by all the software packages. This phenomenon has also been observed in other studies with a decrease in expression when lower stability genes are used for normalisation (58, 59). However, more dramatic differences in expression levels can also be observed with significant alterations of relativities depending on the reference genes used (60). Although patterns of expression were retained, statistical analysis of the differences between the relative expression levels was affected, with a decrease in the significance of the differences between expressions at different time points in planta compared with expression in vitro. This indicates that use of the sub-optimal genes for normalisation may lead to nuanced differences in gene expression being overlooked.

Throughout the manuscript please change qRT-PCR to RT-qPCR. We have changed the abbreviation throughout the manuscript. We have also altered the definition of the abbreviation from ‘real-time quantitative reverse transcription PCR’ to ‘reverse transcription quantitative real-time PCR’, to re-inforce that the RT signifies reverse transcription rather than real-time.

In addition, following comments received after this manuscript was viewed on bioRxiv we have made a couple of further improvements:

Lines 139-140 and 143-144: We have changed the units for measuring temperatures during the pathogenicity tests from growing degree days to thermal time (°C days) as a more appropriate unit for standardisation of growing conditions between experiments. 

Line 443: We have added the comment: There was no amplification of fungal sequences during RT-qPCR analysis when mock-inoculated cDNA was used as template throughout the time course.

---

## [Decision Letter · Decision Letter 1]

2 Nov 2020

Reference genes for gene expression analysis in the fungal pathogen *Neonectria ditissima* and their use demonstrating expression up-regulation of candidate virulence genes

PONE-D-20-24931R1

Dear Dr. Bowen,

We’re pleased to inform you that your manuscript has been judged scientifically suitable for publication and will be formally accepted for publication once it meets all outstanding technical requirements.

Kind regards,

Raffaella Balestrini

Academic Editor

PLOS ONE

Additional Editor Comments (optional):

Reviewers' comments:

Reviewer's Responses to Questions

**Comments to the Author**

1. If the authors have adequately addressed your comments raised in a previous round of review and you feel that this manuscript is now acceptable for publication, you may indicate that here to bypass the “Comments to the Author” section, enter your conflict of interest statement in the “Confidential to Editor” section, and submit your "Accept" recommendation.

Reviewer #1: All comments have been addressed

2. Is the manuscript technically sound, and do the data support the conclusions?

Reviewer #1: (No Response)

3. Has the statistical analysis been performed appropriately and rigorously? 

Reviewer #1: (No Response)

4. Have the authors made all data underlying the findings in their manuscript fully available?

Reviewer #1: (No Response)

5. Is the manuscript presented in an intelligible fashion and written in standard English?

Reviewer #1: (No Response)

6. Review Comments to the Author

Reviewer #1: (No Response)

7. PLOS authors have the option to publish the peer review history of their article (what does this mean?). If published, this will include your full peer review and any attached files.

Reviewer #1: No

---

## [Editor Report · Acceptance letter]

5 Nov 2020

PONE-D-20-24931R1 

Reference genes for gene expression analysis in the fungal pathogen *Neonectria ditissima* and their use demonstrating expression up-regulation of candidate virulence genes 

Dear Dr. Bowen:

I'm pleased to inform you that your manuscript has been deemed suitable for publication in PLOS ONE. Congratulations! Your manuscript is now with our production department. 

Kind regards, 

on behalf of

Dr Raffaella Balestrini 

Academic Editor

PLOS ONE